# Chalcone Derivative Induces Flagellar Disruption and Autophagic Phenotype in *Phytomonas serpens* In Vitro

**DOI:** 10.3390/pathogens12030423

**Published:** 2023-03-07

**Authors:** Tamiris A. C. Santos, Kleiton P. Silva, Gabriella B. Souza, Péricles B. Alves, Rubem F. S. Menna-Barreto, Ricardo Scher, Roberta P. M. Fernandes

**Affiliations:** 1Laboratório de Enzimologia, Departamento de Fisiologia, Universidade Federal de Sergipe, São Cristóvão 49100-000, SE, Brazil; 2Programa de Pós-Graduação em Agricultura e Biodiversidade, Universidade Federal de Sergipe, São Cristóvão 49100-000, SE, Brazil; 3Laboratório de Química, Universidade Federal de Sergipe, São Cristóvão 49100-000, SE, Brazil; 4Laboratório de Biologia Celular, Instituto Oswaldo Cruz, Fundação Oswaldo Cruz, Rio de Janeiro 21040-900, RJ, Brazil; 5Laboratório de Biologia Celular e Imunologia do Câncer e Leishmania, Universidade Federal de Sergipe, São Cristóvão 49100-000, SE, Brazil

**Keywords:** trypanossomatid, phytoparasite, antiprotozoal activity, autophagy, reactive oxygen species

## Abstract

*Phytomonas serpens* is a trypanosomatid phytoparasite, found in a great variety of species, including tomato plants. It is a significant problem for agriculture, causing high economic loss. In order to reduce the vegetal infections, different strategies have been used. The biological activity of molecules obtained from natural sources has been widely investigated to treat trypanosomatids infections. Among these compounds, chalcones have been shown to have anti-parasitic and anti-inflammatory effects, being described as having a remarkable activity on trypanosomatids, especially in *Leishmania* species. Here, we evaluated the antiprotozoal activity of the chalcone derivative (NaF) on *P. serpens* promastigotes, while also assessing its mechanism of action. The results showed that treatment with the derivative NaF for 24 h promotes an important reduction in the parasite proliferation (IC_50_/24 h = 23.6 ± 4.6 µM). At IC_50_/24 h concentration, the compound induced an increase in reactive oxygen species (ROS) production and a shortening of the unique flagellum of the parasites. Electron microscopy evaluation reinforced the flagellar phenotype in treated promastigotes, and a dilated flagellar pocket was frequently observed. The treatment also promoted a prominent autophagic phenotype. An increased number of autophagosomes were detected, presenting different levels of cargo degradation, endoplasmic reticulum profiles surrounding different cellular structures, and the presence of concentric membranar structures inside the mitochondrion. Chalcone derivatives may present an opportunity to develop a treatment for the *P. serpens* infection, as they are easy to synthesize and are low in cost. In order to develop a new product, further studies are still necessary.

## 1. Introduction

Trypanosomatids are flagellated protozoans belonging to the Trypanosomatidae family that infect vertebrates (mainly mammals) and plants, and are transmitted by insects. The genera *Leishmania* and *Trypanosoma* are significant for presenting specimens responsible for the development of neglected tropical diseases, and are therefore of great importance to public health. Furthermore, phytopathogenic species belonging to the genus *Phytomonas* have a global distribution, and have been isolated from different plant tissues such as latex ducts, phloem, fruits, seeds, and flowers of host plants [1,2]. In general, two species have been proved to be pathogenic for plants: *Phytomonas staheli*, the etiological agent of diseases such as phytomonas wilt in coconut trees (*Cocos nucifera*), and *Phytomonas leptovasorum*, the etiological agent of coffee necrosis. The species *Phytomonas serpens* causes damage to tomato plants, including loss of viability in ripe fruits with color changes from reddish to golden [3]. Taking into account the economic importance of these crops, diseases caused by phytomonas parasites represent a significant problem to agriculture, leading to high economic loss.

Nowadays, there is no effective treatment for these diseases, and the only control alternative is based on eradicating the infected plants [2]. The screening of new molecules is hampered by the difficulties in isolating the phytopathogenic species such as *P. staheli* and *P. leptovasorum*, and the difficulty of in vitro cultivation of these parasites. In this context, *P. serpens* emerges as an important model, easily isolated from infected tomatoes and cultivated in the laboratory. Moreover, genetic and biochemical similarities between nonpathogenic and pathogenic genera of trypanosomatides have been described [4,5]. The genomic analysis of *P. serpens* revealed similarities with species of the genus *Leishmania* and *Trypanosoma*, including the presence of homologue genes related to virulence such as Gp63 and cruzipain [5,6]. Thus, similar molecular machineries are present in both clinically relevant trypanosomatids and *P. serpens*, making this phytoparasite a good model for in vitro studies [7].

Exploring natural products has been a good strategy to search for new candidates for the development of new drugs against protozoa, including trypanosomatids. The literature points out a wide variety of essential oils [8,9], plant extracts [10,11], and isolated compounds [12,13,14] that show anti-trypanosomatids activity. Among the main classes of active natural products are alkaloids, quinones, terpenes and phenolic compounds [15]. Recently, chalcone (1,3-diaryl-2-propane-1-one), a phenolic precursor made up of an α β-unsaturated carbonyl system that joins two aromatic rings [16], and its derivatives have stood out [17,18,19,20,21]. The inhibitory activity of a large amount of chalcone derivatives has already been well-described against *Leishmania*, *T. cruzi* and *T. brucei* [22,23,24]. In these parasites, chalcones and its derivatives mainly affect the mitochondrion, causing structural alterations in this organelle, as well as cytoplasmic and nuclear changes including vacuolization and DNA fragmentation [25,26]. In adition, chalcones also inhibit enzymes related to aerobic metabolism and can induce ROS production, altering plasma membrane fluidity [27,28].

In *Phytomonas* spp., the activity of chalcones and derivatives was first evaluated by our group recently, being the compound (E)-1-phenyl-3-αnaphthylprop-2-en-1-one (NaF), which is the most active [29]. Here, we further investigated the derivative NaF mode of action by transmission and scanning electron microscopy approaches to identify the target organelles and/or parasite structures affected, and the complementary analysis of ROS generation and verification of plasma membrane integrity.

## 2. Materials and Methods

### 2.1. Reagents

Dimethylsulfoxide (DMSO), fetal bovine serum (FBS), Schneider medium, streptomycin, hydrogen peroxide (H_2_O_2_), H_2_DCFDA, propidium iodide (PI), glutaraldehyde (GA), osmium tetroxide (OsO_4_), dithiothreitol (DTT), methanol, and cacodylic acid were purchased from Sigma Aldrich (St Louis, MO, USA). All other reagents were of analytical grade or better. PolyBed 812 resin was purchased from Polysciences, Inc. (Warrington, USA) and Panoptic Fast Kit from Laborclin (Pinhais, Brazil).

### 2.2. NaF Synthesis

The chalcone derivative (E)-1-phenyl-3-αnaphthylprop-2-en-1-one (NaF) was synthesized through aldol condensation (Claisen–Schmidt condensation). Identification and characterization were performed by absorption spectrometry in the infrared region, gas chromatography coupled with mass spectrometry and flame ionization detector, ^1^H and ^13^C nuclear magnetic resonance spectroscopy, and high-resolution mass spectrometry, as previously described [29] (Appendix A). The dissolution of NaF was carried out with 1% DMSO to evaluate the biological effects on the trypanosomatid.

### 2.3. Parasites Culture and the Derivative NaF 

*P. serpens* isolate (strain 9T) was obtained from the trypanosomatids collection of the Oswaldo Cruz Foundation (Fiocruz, Brazil). Promastigotes were maintained in Schneider medium supplemented with 10% FBS and 1% streptomycin and incubated at 25 °C. Cells were replicated every seven days. To determine the IC_50_/24 h, a culture of phytomonas was initiated by inoculating 5 × 10^6^ cells/well in the logarithmic growth phase in 12-well plates, with serial dilutions of NaF ranging from 6.0 to 96.0 µM. After 24 h, parasites were counted using a Neubauer chamber and the IC_50_/24 h was calculated by sigmoidal regression.

### 2.4. Analysis of Plasma Membrane Integrity

Promastigotes (1 × 10^6^/mL) were cultured in the absence and presence of concentrations (6–96 µM) of the derivative NaF for 24 h at 25 °C. After the incubation period, parasites were resuspended (1 × 10^7^/mL) in phosphate buffer saline (PBS) and incubated with PI (20 µM) for 30 min in the dark. The positive control was performed with cells exposed to 80 °C for 10 min. Fluorescence was measured at the wavelength (493/632 nm) using a Hybrid Multi-Mode Reader Synergy H1 (Santa Clara, CA, USA), and analysis was run at Gen52.06 software (Santa Clara, CA, USA).

### 2.5. Analysis of Reactive Oxygen Species (ROS) Production

Promastigotes (1 × 10^6^ parasites/mL) were incubated in the absence and presence of NaF (6, 12 and 24 µM) for 24 h at 25 °C. Then, parasites were resuspended in PBS (1 × 10^7^/mL) and labelled with H_2_DCFDA (20 µM) for 30 min. Simultaneously, the incubation with 750 µM of H_2_O_2_ for 30 min was employed as a positive control. Fluorescence was measured at wavelength (492/527 nm) using a Hybrid Multi-Mode Reader Synergy H1 (Santa Clara, USA), and analysis was run at Gen52.06 software (Santa Clara, CA, USA).

### 2.6. Ultrastructural Analysis

Promastigotes (1 × 10^7^ parasites/mL) were treated with NaF for 24 h in Schneider medium at 25 °C. Afterward, the cells were fixed with 2.5% glutaraldehyde in 0.1 M Na-cacodylate buffer (pH 7.2) at room temperature for 40 min at 25 °C, and post-fixed with a solution of 1% OsO_4_, 0.8% potassium ferricyanide, and 2.5 mM CaCl_2_ in the same buffer for 20 min at 25 °C. The samples were dehydrated in an ascending acetone series and embedded in PolyBed 812 resin. Ultrathin sections were stained with uranyl acetate and lead citrate, and examined in a Jeol 1011 transmission electron microscope (Tokyo, Japan). Alternatively, samples were also dehydrated in an ascending ethanol series and dried by the critical point method with CO_2_, mounted on aluminum stubs, coated with a 20 nm thick gold layer, and examined in Jeol JSM6390LV scanning electron microscope (Tokyo, Japan). All ultrastructural analysis were performed at Plataforma de Microscopia Eletrônica, Instituto Oswaldo Cruz, FIOCRUZ.

### 2.7. Effect of NaF on the Promastigote Flagellar Morphology

Promastigotes (1 × 10^6^ cells/mL) were cultured in the absence of NaF and the presence of 24 µM of NaF for 24 h. Cells were fixed in methanol and stained with a fast panoptic kit and evaluated under a light microscope. Quantification was performed by counting the number of flagellated promastigotes in relation to the total number of promastigotes. At least 150 parasites were evaluated per experimental condition. 

### 2.8. Statistical Analysis

All experiments were performed at least in biological triplicate. Data were expressed as mean ± standard error. Statistical analysis was performed using the GraphPad Prism 7.01 programusing Two-Way ANOVA and means were compared by t-test, Dunnett, being considered different *p* values < 0.05, (* *p* < 0.05, ** *p* < 0.01, *** *p* < 0.001, **** *p* < 0.0001).

## 3. Results

### 3.1. NaF Affects P. serpens Promastigotes Proliferation

After 24 h, all evaluated concentrations of NaF promoted a significant reduction in parasite proliferation. IC_50_/24 h value calculated was 23.6 ± 4.6 µM. The concentrations of 48 and 96 µM led to a growth reduction of 66.9% and 75.4%, respectively (Figure 1).

### 3.2. The Compound NaF Induces an Increase in ROS Production without Affecting Plasma Membrane Integrity

The probe H2DCFDA was employed to assess ROS levels in treated parasites. NaF promoted a significant 2-fold increase in H2DCFDA fluorescence at 24 µM. Despite not being statistical, lower concentrations (6 and 12 µM) also led to a tendency of an increase in the marker fluorescence (Figure 2). On the other hand, parasites exposed to the concentrations up to 4-fold IC_50_/24 h (96 µM) show no PI labeling 

### 3.3. The Derivative NaF Induces Flagellarloss and/or Shortening

To assess the effect of the compound NaF on the parasite’s morphology, scanning microscopy analysis was performed. A high number of treated parasites without an apparent flagellum was detected (Figure 3). In order to reinforce these qualitative data, the frequency of this phenotype was assessed by light microscopy, confirming the ultrastructural findings. Almost 3-fold more promastigotes with the flagellar shortening was observed after the treatment (Figure 4).

### 3.4. The Derivative NaF Promoted the Appearance of Autophagic Phenotype in P. serpens

Transmission electron microscopy analysis revealed a recurrent autophagic phenotype in treated promastigotes, including the increase of the number of autophagosomes, detected in different levels of cargo degradation, the well-developed endoplasmic reticulum profiles, and the formation of cytosolic and mitochondrial concentric membranar structures. Flagellar pocket swelling with innumerous concentric membranar structures inside could be also observed after the treatment (Figure 5).

## 4. Discussion

Agricultural diseases such as phloem necrosis in coffee and phytomonas wilt in coconut trees are caused by parasites of the *Phytomonas* genus, leading to relevant economic losses. In 2001, it was recorded that 43% of coconut plantations in the state of Amazonas in Brazil were affected with phytomonas wilt, and several plantations were decimated [30]. Coffee phloem necrosis represents a potential risk in Brazil, since this country stands out in the market as a coffee producer. Unfortunately, there is no treatment for these diseases, and the control strategies depend on eradicating infected plants [2]. Additionaly, the emergence of protozoa subpopulations resistant to the commercial drugs strongly indicates the requirement of multidisciplinary efforts for the development of alternative compounds with anti-trypanosomatids activity. In the early steps of this continuous search, cellular, molecular and biochemical information on parasites must be compiled, as well as the identification of potential targets of novel prototypes, and critical checkpoints for the characterization of drug safety and specificity. In the agricultural scenario, pharmacological studies aiming to screen alternative compounds for the treatment are scarce, especially due to the limitation of in vitro multiplication of *P. leptovasorum* and *P. staheli*. However, *P. serpens* can be easily cultivated in laboratorial conditions, being a promising model. Following the example of other diseases, approaches including high throughput screenings analysis in *Phytomonas* spp. would be crucial to accelerate the discovery of novel candidates for the treatment. Up until now, few mechanistic studies were published in this area, with challenges yet to be overcome.

The anti-*P. serpens* activity of natural products such as *Varroniacuras savica* and *Lantana camara* essential oils, as well as alkaloids including tomatine and tomatidine, have been reported [9,11,13]. Chalcones have been considered promising molecules due to their broad range of biological activities, multi-target action, simple chemistry and low toxicity [31]. The effect of chalcones and their derivatives has already been described for other protozoa [20,21,26,32,33]. In a previous screening, our group selected a chalcone derivative named NaF based on its activity in reducing *P. serpens* viability in vitro [29]. Such a derivative is a compound that presents a naphthyl group in the ring B, showing the described antibacterial [34], antitumoral [35], nematicidal [36], and spermicidal [37] activities. However, no reports about its anti-trypanosomatids effect and its mechanism of action have been published up until now. Here, the IC_50_/ 24 h value calculated for promastigotes was in the range of 24 µM, with this molecule considered moderately active, following Upegui and collaborators categorization [38]. Despite this interesting activity, it is crucial that the improvement of the derivative, and designing novel substitutions that affect the electrophilicity of the α,β-unsaturated carbonyl moiety, also influence the binding ability of the compound NaF, and consequently improve its biological effect.

As is well-known, molecular and biochemical tools for the fast and efficient identification of drug targets are still lacking. In order to fill this gap, ultrastructural analysis has been frequently used, at least for the first assessment of primary targets affected by drugs in parasites including trypanosomatids, allowing inferences about their mechanism of action [39]. Aiming to further investigate the mode of action of the derivative NaF, ultrastructural, flow cytometric and fluorimetric analysis were performed on *P. serpens*. Surprisingly, no loss of plasma membrane integrity was detected in the range of concentrations employed (6–96 µM). At the highest dose tested, 75% of the parasites died, but no plasma membrane disruption was observed. It was previously observed in chalcones-treated *Leishmania* (Viannia) *braziliensis* [26], excluding the plasma membrane as a primary target of this compound in *P. serpens*. 

In the context of the natural products’ mode of action, oxidative stress figures as one of the most recurrent processes directly involved in their biological effect. The increase in ROS production as a result of chalcones treatment was already reported in trypanosomatids [22,23,24,40]. Here, the compound NaF at IC_50_/24 h concentration induced a remarkable increase in ROS levels in *P. serpens* promastigotes, similar to that previously observed in *T. cru*z*i* and *L. amazonensis* treated with other plant secondary metabolites, sesquiterpenoids and flavonoids*,* respectively [41,42]. Interestingly, in these studies the increased ROS production was associated with mitochondrial damage, evidenced by the organelle swelling and reduction in the mitochondrial membrane potential. More recently, our group demonstrated the crucial role of mitochondrial ROS generation for the trypanocidal activity of naphthoimidazoles derived from beta-lapachone isolated from *Tabebuia* trees [43,44]. These previous data pointed to a close relation between mitochondrial injury (primary or not) and the increase of ROS generation. In the present study, ROS production induced by the derivative NaF in *P. serpens* could not be related to the impairment of mitochondrial ultrastructure. On the other hand, an increase in ROS levels without electron microscopy alterations in this organelle has already been reported in *T. cruzi* trypomastigotes treated with a triazolic naphthoquinone [45], revealing that the chemical structure of the compound directly influences its capacity to reach and consequently affect the mitochondrion. Therefore, we can consider that the ROS induced by NaF in *P. serpens* is accumulated in the parasite by a mitochondrion-independent pathway, but further analysis must be performed to better characterize the molecular mechanism involved.

As previously stated, *Phytomonas* spp. are trypanosomatids found in plant tissues, which have carbohydrates as their main source of energy [46]. ROS derived from normal aerobic metabolism is neutralized by a number of antioxidant enzymes, including superoxide dismutase (SOD), glutathione peroxidases (GPxs), and thioredoxin (Trx), as well as the non-enzymatic antioxidants that collectively reduce the oxidative state [46]. In trypanosomatids, especially in extracellular forms, the repertoire of antioxidant enzymes is reduced due to the loss of coding genes for several of these enzymes [47,48]. Once in Phytomonas, the antioxidant defense is based on the reduced form of the thiol trypanothione, a homodimer of glutathione (GSH) [6,49]. ROS accumulation observed after the treatment with the compound NaF could be explained by its reaction with trypanothione. As a chalcone derivative, NaF presents the α,β-unsaturated carbonyl functional moiety characterized as the Michael acceptor. This electrophilic agent participates in the covalent bond with thiol and other similar nucleophiles through the Michael addiction [50]. In this way, the NaF α,β-unsaturated bonds, non-enzymatically, to the thiol group of GSH of trypanothione, thereby compromising the redox pathway trypanothione-dependent. Michael addiction involving chalcone derivative and glutathione thiols have already being reported in human cancer cells, resulting in GSH depletion and ROS elevation [51,52]. Furthermore, recent in vitro and in silico studies suggest that chalcones are able to bind in crucial sites of trypanothione reductase (TR) and tryparedoxin peroxidase (TXNPx), affecting their activity which leads to increased ROS levels [22,23,24,40]. Thus, ROS accumulation observed in *P. serpens* after the treatment with the chalcone NaF could be derived from the parasite’s redox unbalance, rather than causing a secondary mitochondrial dysfunction. It is evident that the oxidative stress triggered by this compound plays a role in *P. serpens* viability, and it will be investigated by our group in the near future.

Moreover, the chalcone derivative NaF also induced some morphological alterations. Scanning electron and light microscopy analysis pointed to a significantly higher number of non-flagellated parasites or shortened flagellum in protozoa exposed to IC_50_/24 h. In transmission electron microscopy, a disruption in the flagellar pocket was commonly observed, corroborating these data. Changes in the trypanosomatids flagellum have been reported after treatment with natural or synthetic compounds [53]. *Leishmania (Leishmania) mexicana* promastigotes treated with a chalcone derivative presented a short flagella [32]. Obviously, the flagellar shortening compromises the parasite motility, and once the flagellar motility demands high ATP amounts, it would be reasonable to assume that a mitochondrial impairment could be involved. However, only discrete damage was observed in the organelle by transmission electron microscopy, suggestive of the preserved mitochondrial architecture. 

Despite the fact that there is no molecular evidence to support this hypothesis up until now, the present aromatic moieties conferes to the derivative NaF a planar chemical structure, increasing the possibility of nucleic acids intercalation, even allosterically, as was previously suggested for naphthoimidazoles [54,55]. Such interaction with macromolecules such as DNA could interfere somehow in the transcriptional steps, affecting gene expression and explaining the appearance of the unexpected phenotype. Despite the robust morphological data presented here, further molecular and biochemical analysis must be performed to better evaluate the molecular mechanisms involved. In the same direction, the formation of flagellar membrane blebbing could represent a parasite’s attempt to eliminate an abnormal structure and/or macromolecule in the flagellar region by a shedding process.

Inummerous reports correlated drug activity and cell death induction, many of them based on morphological data, mainly transmission electron microscopy findings. On the other hand, the existence of programmed cell death in protozoa has been highly debated, due to the lack of critical information about regulatory processes and the biochemical and molecular events involved. Based on the absence of the identification of molecular executioners, protozoan parasites’ cell death is still considered to be an unregulated process in the literature [56]. The most recurrent ultrastructural damage observed in treated parasites was the autophagy-related phenotype. Autophagy is a conserved pathway presented in all eukaryotic cells, responsible for remodelling cellular structures. It is well-known that loss of homeostasis can result in an increase in autophagic levels and subsequent cell death [56]. Autophagic features such as prominent endoplasmic reticulum profiles and concentric membrane structures in mitochondria, in addition to the formation of autophagosomes, were recurrently observed. Such features have been described in different trypanosomatids under stress conditions [57], including treatment with drugs such as sesquiterpenoids, naphthoquinones or clomipramine [41,58,59]. In fact, such a phenotype has been extensively described in different species of protozoa treated with a great variety of drugs. The most recurrent autophagic features are the appearance of myelin-like figures, exacerbation of the autophagosomes number as well as the endoplasmic reticulum profiles close to distinct cellular structures. Morphological approaches suggest the involvement of autophagy in the cell death process triggered by the compounds, despite the fact that no regulatory events have been postulated [39]. The ultrastructural evidences clearly pointed to the autophagy as the least part of the mechanism of action of this derivative. On the other hand, the description of this pathway in parasites treated with all classes of compounds raise the hypothesis of an unspecific effect. Once distinct drugs interfere in all kind of molecular pathways, a non-selective autophagy could take place in order to turn over the damaged cellular structures, acting as a survival machinery [39]. As has been proposed for other compounds, NaF probably presents an specific unknown anti-*P. serpens* action initially assessed here. However, the treatment culminates in an autophagic phenotype, a parasite survival mechanism. Such a hypothesis explains why this phenotype is frequently observed in almost all protozoan parasites after treatment with different classes of drugs, regardless of the mode of action involved [39,57].

## 5. Conclusions

Our study presents an initial overview of the action mode of chalcone derivative NaF that is easily synthetized and shows a strong antiproliferative effect on *P. serpens* promastigotes. Besides affecting the oxidative homeostasis, NaF treatment resulted in flagellar impairment and an autophagic phenotype. Together, these results demontrate that NaF exhibits in vitro activity against *P. serpens*, and can represent a good starting point for the development of a new treatment to control plant diseases caused by *Phytomonas* spp.

## Figures and Tables

**Figure 1 pathogens-12-00423-f001:**
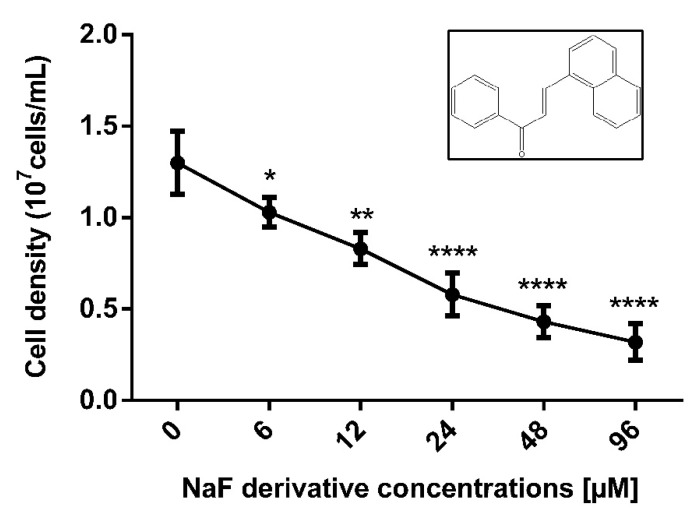
Effect of the derivative NaF on *P. serpens* promastigotes proliferation. Promastigotes were cultured in presence of NaF concentrations ranging from 6 to 96 µM. Parasite proliferation was determined by quantification in a Neubauer chamber after 24 h. The experiments were performed in triplicate with three independent biological experiments. Data show mean ± SE. Statistical differences were analyzed by ANOVA and Dunnett’s test. Significant difference was observed in relation to untreated control (* *p* < 0.05, ** *p* < 0.01, **** *p* < 0.0001)**.** Box shows the compound chemical structure.

**Figure 2 pathogens-12-00423-f002:**
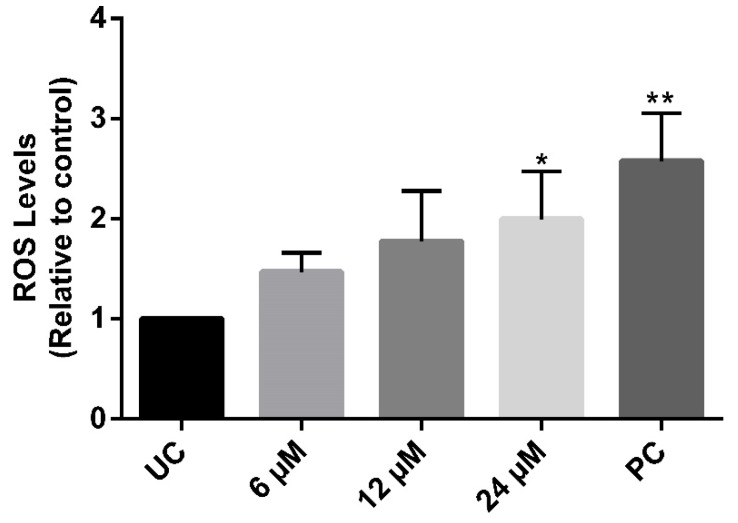
ROS levels in *P. serpens* promastigotes treated with the compound NaF. Parasites were treated with 6, 12, and 24 µM of NaF for 24 h and stained with H_2_DCFDA for 30 min at 25 °C. Cells incubated with hydrogen peroxide used as a positive control (PC). Fluorescence intensity was determined by fluorimetry. The fluorescence mean data were normalized by the untreated control (UC). Bars represent mean ± S.E. The experiments were performed in triplicate with three biological independent experiments. Statistical differences were analyzed by ANOVA and Dunnett’s test. Significant difference was observed in relation to untreated control (* *p* < 0.05, ** *p* < 0.01).

**Figure 3 pathogens-12-00423-f003:**
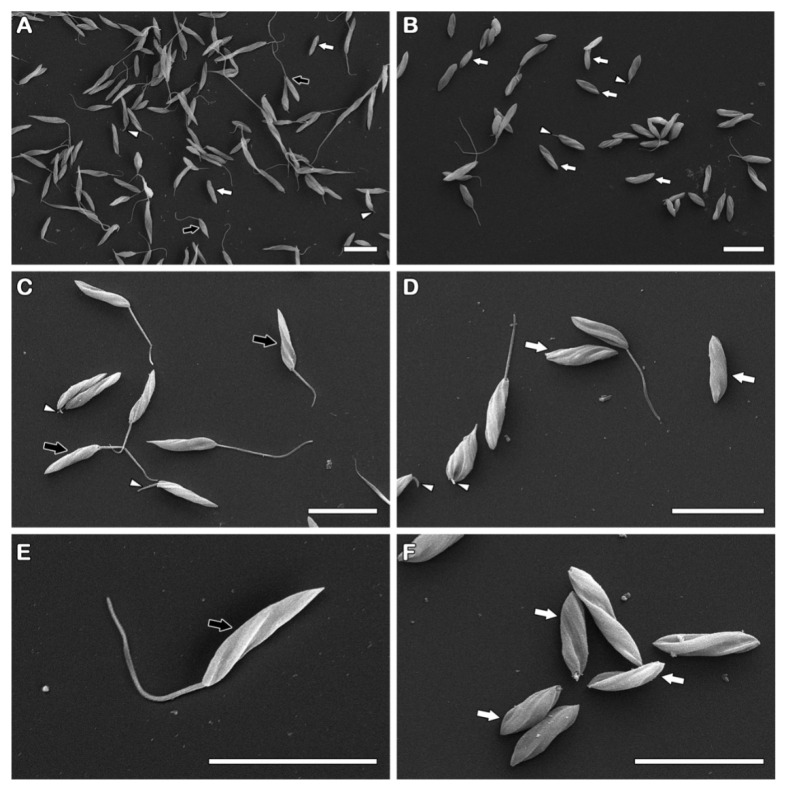
Flagellar loss and/or shortening was induced by the treatment with the derivative NaF in promastigote forms of *P. serpens*. (**A**,**C**,**E**) Control, untreated cells; (**B**,**D**,**F**) Promastigotes treated with 24 µM of the compound. (**A**–**F**) The parasites showed typical morphology (black arrows) or flagellar shortening (white arrowheads). Promastigotes without the flagellum (white arrows) were easily found after treatment with 24 µM. Bars = 10 µm.

**Figure 4 pathogens-12-00423-f004:**
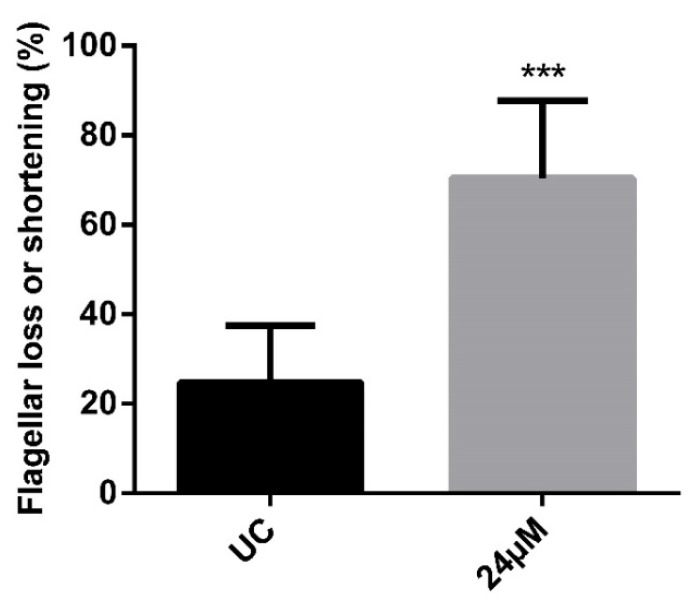
Frequency of *P. serpens* promastigotes showing flagellar loss or shortening. Parasites were treated or not with 24 µM of the compound NaF for 24 h. Cells were stained with panotype dye and counting in light microscopy. Bars represent mean ± S.E. The experiments were performed in triplicate with three independent biological experiments. Statistical differences were analyzed by *t* test. Significant difference was observed in relation to untreated control (UC) (*** *p* < 0.001).

**Figure 5 pathogens-12-00423-f005:**
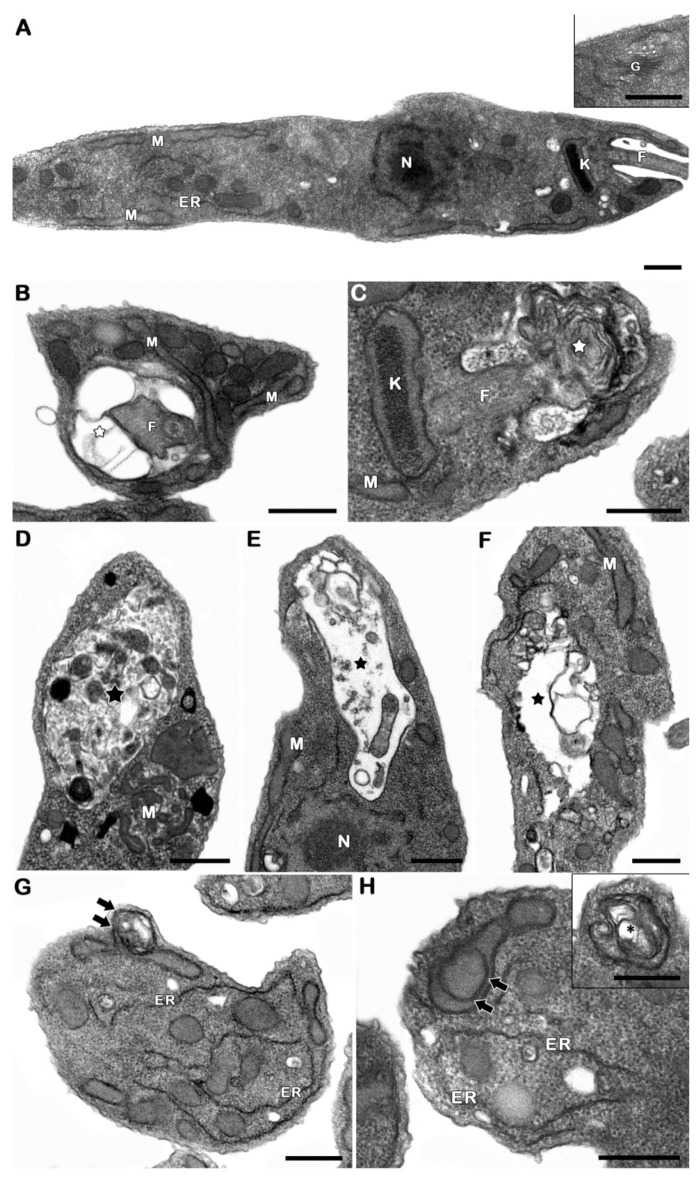
Ultrastructural analysis showed an autophagic phenotype in *P. serpens* promastigote forms treated with 24 µM of the derivative NaF (IC_50_/24 h). (**A**) Untreated parasites showed normal morphology. (**B**–**F**) Treatment with 24 µM NaF induced swelling in the flagellar pocket. Within this structure were concentric membrane structures (white stars), as well as the formation of autophagosomes with variable content (black stars). (**G**–**H**) Parasites treated with 24 µM also showed prominent endoplasmic reticulum profiles, in addition to the recurrent presence of concentric membrane structures within the mitochondrion (black arrows) and in the cytosol (black asterisks in detail). N: nucleus; M: Mitochondria; K: Kinetoplast; F: flagellum; ER: Endoplasmicreticulum; G: Golgi. Bars = 5 μm.

## Data Availability

Not applicable.

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
