# Peer review of "Chalcone Derivative Induces Flagellar Disruption and Autophagic Phenotype in Phytomonas serpens In Vitro"

_pathogens, 2023, doi:10.3390/pathogens12030423_

Round 1

Reviewer 1 Report

The manuscript is very interesting and fits well into the special issue of Pathogens journal. However, the manuscript must be carefully checked taking into account the English grammar and the several typographical errors. In this context, some of the mistakes are highlighted, but not all of them.

(i) Title

- The reviewer strongly suggests changing NaF from the title, since it can be confused with sodium fluoride. Please, add the full name of the selected test compound or another abbreviation.

(ii) Abstract

- Since the trypanosomatids have a unique flagellum, change the sentence “the shortening of the parasites flagella” to “the shortening of the unique flagellum of the parasites”

- NaF abbreviation, please, see the reviewer's suggestion regarding this abbreviation in the title.

(iii) Introduction

- The sentence in line 40 contains many errors and requires attention.  Change “Furthermore, the genus Phytomonas include phytopatogenic species that have a global distribution and have been isolated from different plant tissues, such as latex ducts, phloem, fruits, seeds, and flowers of host plants.” To “Furthermore, phytopathogenic species belonging to the genus Phytomonas have a global distribution and have been isolated from different plant tissues, such as latex ducts, phloem, fruits, seeds, and flowers of host plants.”

- Line 45, change “The specie Phytomonas serpens” to “The species Phytomonas serpens”

- Lines 59-60, the sentence “Thus, P. serpens has been considered a phytoparasite of medical importance” is not correct, since P. serpens does not have medical relevance.

- Line 69, change “Inhibitory activity of a large amount of chalcone derivatives have” to “Inhibitory activity of a large amount of chalcone derivatives has”

- Line 76, “spp.” is not in italic.

- The last sentence of the introduction “Here, we further investigated NaF mode of action, analyzing its effect by ultrastructural and fluorimetric analysis.” should be better detailed, because it is vague. Moreover, add “the” before NaF.

(iv) Methods

- Change “bovine fetal serum (BFS)” to “fetal bovine serum (FBS)”

- Change the subtitle “Parasites and NaF IC50/24h calculation”, because it is confusing.

- Line 17, many words are joined together

(v) Results

- The authors said “Using the probe H2DCFDA, the ROS levels were assessed after the treatment. NaF promoted a significant increase at 24 μM, in range of 2-fold. Lower concentrations (6 and 12 μM) also led to a tendency of increase, but it was not statistical (Figure 2).” These sentences should be rewritten.

- The authors said “Surprisingly, parasites exposed to the concentrations up to 4-fold IC50/24h (96 μM) show no PI labeling (data not shown).” Please, why was it a surprise? Did the authors detect the ROS level after 48 h, for example, in order to check if the ROS level will be increased or decreased? The test compound can increase the ROS level but also it can increase the antioxidant responses of the parasites; do the authors check it?

- The parasites’ flagellum was unquestionably affected by the NaF treatment. Did the authors evaluate the mitochondrial functionality? The reviewer is asking because flagellar movement is driven by mitochondrial-derived energy.

Author Response

The manuscript is very interesting and fits well into the special issue of Pathogens journal. However, the manuscript must be carefully checked taking into account the English grammar and the several typographical errors. In this context, some of the mistakes are highlighted, but not all of them.

 (i) Title

- The reviewer strongly suggests changing NaF from the title, since it can be confused with sodium fluoride. Please, add the full name of the selected test compound or another abbreviation.

(ii) Abstract

- Since the trypanosomatids have a unique flagellum, change the sentence “the shortening of the parasites flagella” to “the shortening of the unique flagellum of the parasites”

- NaF abbreviation, please, see the reviewer's suggestion regarding this abbreviation in the title.

  Answer: Thanks for your comment. We completely agree with you and reviewer#2 too. Following the recommendation of both of you, we delete NaF from the title, and included in the whole the “derivative” or “compound” before NaF to avoid misinterpretation. In relation to the origin of the abbreviation, this compound is synthetized from naphthalene that in Portuguese is written “naftaleno”. It is a “nick name” that we used in lab… If you think that it is still confused, we can replace by other abbreviation.   The sentence “the shortening of the unique flagellum of the parasites” replaced the old one.

 (iii) Introduction

- The sentence in line 40 contains many errors and requires attention.  Change “Furthermore, the genus Phytomonas include phytopatogenic species that have a global distribution and have been isolated from different plant tissues, such as latex ducts, phloem, fruits, seeds, and flowers of host plants.” To “Furthermore, phytopathogenic species belonging to the genus Phytomonas have a global distribution and have been isolated from different plant tissues, such as latex ducts, phloem, fruits, seeds, and flowers of host plants.”

 - Line 45, change “The specie Phytomonas serpens” to “The species Phytomonas serpens”

- Lines 59-60, the sentence “Thus, P. serpens has been considered a phytoparasite of medical importance” is not correct, since P. serpens does not have medical relevance.

- Line 69, change “Inhibitory activity of a large amount of chalcone derivatives have” to “Inhibitory activity of a large amount of chalcone derivatives has”

- Line 76, “spp.” is not in italic.

- The last sentence of the introduction “Here, we further investigated NaF mode of action, analyzing its effect by ultrastructural and fluorimetric analysis.” should be better detailed, because it is vague. Moreover, add “the” before NaF.

 Answer: All mistakes were corrected; In lines 59-60, a new sentence was written: “Thus, similar molecular machineries are present in both clinical relevant trypanosomatids and P. serpens, making this phytoparasite a good model for in vitro studies” The last sentence of the introduction was rewritten: “we further investigated the derivative NaF mode of action by transmission and scanning electron microscopy approaches to identify target organelles and/or parasite structures affected, and complementary analysis of ROS generation and verification of plasma membrane integrity”

 (iv) Methods

 - Change “bovine fetal serum (BFS)” to “fetal bovine serum (FBS)”

- Change the subtitle “Parasites and NaF IC50/24h calculation”, because it is confusing.

- Line 17, many words are joined together

 Answer: All mistakes were corrected; the new subtitle is “Parasites culture and the derivative NaF”; the whole text was carefully revised and all joined words were separated.

  (v) Results

- The authors said “Using the probe H2DCFDA, the ROS levels were assessed after the treatment. NaF promoted a significant increase at 24 μM, in range of 2-fold. Lower concentrations (6 and 12 μM) also led to a tendency of increase, but it was not statistical (Figure 2).” These sentences should be rewritten.

 Answer: We completely agree with the reviewer, the sentence was truncated. The new sentences were written: “The probe H2DCFDA was employed to assess ROS levels in treated parasites. NaF promoted a significant 2-fold increase in H2DCFDA fluorescence at 24 µM. Despite not statistical, lower concentrations (6 and 12 µM) also led to a tendency of increase in the marker fluorescence (Figure 2).”

 - The authors said “Surprisingly, parasites exposed to the concentrations up to 4-fold IC50/24h (96 μM) show no PI labeling (data not shown).” Please, why was it a surprise? Did the authors detect the ROS level after 48 h, for example, in order to check if the ROS level will be increased or decreased? The test compound can increase the ROS level but also it can increase the antioxidant responses of the parasites; do the authors check it?

 Answer: Thanks for your comment. The answer for your question is no, we did not analyze ROS levels at other times of treatment such as 48h nor antioxidant responses at any time. Our motivation to focus on only time for all our analysis is to try correlate all our data. The idea was to use a sublethal dose (up to IC50) to transmission electron microscopy assays, trying to identify primary targets. It would be interesting to assess the effect of preincubation with antioxidants such as NAC or urate, as well as further mitochondrial and enzymatic analysis to better comprehend the mechanisms. We hope we can do something in this direction in near future. Anyway, for sure the word “surprisingly” was not a good choice. We replaced it by “On the other hand”.

 - The parasites’ flagellum was unquestionably affected by the NaF treatment. Did the authors evaluate the mitochondrial functionality? The reviewer is asking because flagellar movement is driven by mitochondrial-derived energy.

 Answer: Thanks for the words. Actually, we tried to assess the mitochondrial membrane potential (DYm) by the rhodamine 123 labeling in fluorescence approaches. Due to the unspecific nature of the labeling (the marker binds to H+, whatever it is), it is not a strong information by ilself; we also tried to use the ionophore CCCP as a crucial positive control, dissipating DYm. Unfortunately, the ionophore did not work well, and we preferred not include these fragile data. By the way, our strong ultrastructural and reproductible data pointed to a discrete effect on this organelle, suggestive of no remarkable biochemical alteration in the mitochondrion. Our group recently revised the mitochondrion of trypanosomatids as a drug target, describing a correlation between ultrastructural and biochemical findings (Mem Inst Oswaldo Cruz. 2022, doi: 10.1590/0074-02760210379; Bioenerg Commun 2022, doi 10.26124/bec:2022-0020). Interestingly, in an old paper from our group, we described that naphthoimidazoles affected T. cruzi DYm, but not the parasite motility by video fluorescence microscopy (J Antimicrob Chemother 2005; doi: 10.1093/jac/dki403; Parasitol Res 2007; doi: 10.1007/s00436-007-0556-1).

Reviewer 2 Report

The article tests the effect of a chalcone derivative (NaF) as an anti-protozoan agent against the phytopathogen Phytomonas serpens. Therefore, the effect of different concentrations of NaF on proliferation, production of reactive species of oxygen and morphological features (such as shortening of flagellum and presence of autophagosomes) were assessed and described. They found that NaF is able to significantly reduce P. serpens proliferation (IC 50/24 h of 24 uM). On the other hand, ROS production induced by NaF seems not to be related to the impairment of mitochondrial ultrastructure. Overall, the work was well performed using appropriate methods and the results are sound. In my opinion the paper should be accepted for publication.

Minor points:

-          The chalcone derivative used was the (E)-1-phenyl-3-αnaphthylprop-2-en-1-one  that was named as “NaF” . Although the abbreviation is mentioned in the text, it was not clear for me its origin. I went to a previous paper from the same group (“Synthesis of chalcone derivatives by Claisen Schmidt condensation and in vitro analyses of

-          their antiprotozoal activities) that describes the synthesis of several chalcone derivatives and I could not find the any of the referred as NaF (it seems that they used in the submitted paper the Compound C-5). I would ask the authors if they agree to try to clarify this point since NaF is also the chemical formula of sodium fluoride.

-          Line 45 – replace “specie” for “species”

-          Line 103. Authors mention the use of NaF derivatives (plural) but it seems that only one compound was used.

-          Line 238 – replace “higher” for “highest”. Also, try to rephrase the sentence from lines 237-240 because it is confusing.

Author Response

The article tests the effect of a chalcone derivative (NaF) as an anti-protozoan agent against the phytopathogen Phytomonas serpens. Therefore, the effect of different concentrations of NaF on proliferation, production of reactive species of oxygen and morphological features (such as shortening of flagellum and presence of autophagosomes) were assessed and described. They found that NaF is able to significantly reduce P. serpens proliferation (IC 50/24 h of 24 uM). On the other hand, ROS production induced by NaF seems not to be related to the impairment of mitochondrial ultrastructure. Overall, the work was well performed using appropriate methods and the results are sound. In my opinion the paper should be accepted for publication.

 Minor points:

-          The chalcone derivative used was the (E)-1-phenyl-3-αnaphthylprop-2-en-1-one  that was named as “NaF” . Although the abbreviation is mentioned in the text, it was not clear for me its origin. I went to a previous paper from the same group (“Synthesis of chalcone derivatives by Claisen Schmidt condensation and in vitro analyses of their antiprotozoal activities) that describes the synthesis of several chalcone derivatives and I could not find the any of the referred as NaF (it seems that they used in the submitted paper the Compound C-5). I would ask the authors if they agree to try to clarify this point since NaF is also the chemical formula of sodium fluoride.

 Answer: Thanks for your comment. We completely agree with you and reviewer#1 too. Following the recommendation of both of you, we delete NaF from the title, and included in the whole the “derivative” or “compound” before NaF to avoid misinterpretation. In relation to the origin of the abbreviation, this compound is synthetized from naphthalene that in Portuguese is written “naftaleno”. It is a “nick name” that we used in lab… If you think that it is still confused, we can replace by other abbreviation. 

-          Line 45 – replace “specie” for “species”

-          Line 103. Authors mention the use of NaF derivatives (plural) but it seems that only one compound was used.

-          Line 238 – replace “higher” for “highest”. Also, try to rephrase the sentence from lines 237-240 because it is confusing.

 Answer: All mistakes were corrected.  As suggested, the sentence was rewritten: “At the highest dose tested, 75% of parasites died, but no plasma membrane disruption was observed, as it was previously observed in chalcones-treated Leishmania”.

Reviewer 3 Report

The study is an extension of some previous work of the authors on the activity of chalconess and delivatives in Phytomonas spp., where they first evaluated that the compound (E)-1-phenyl-3-αnaphthylprop-2-en-1-one (NaF) is the most active. This study reports the results of further investigated NaF mode of action, analyzing the effect by ultrastructural and fluorimetric analysis. The result presents an initial overview of the action mode of chalcone derivative NaF that is easily synthetized and shows a strong antiproliferative effect on the promastigotes of P. serpens; suggesting a good start point for the development of a new treatment to control plant diseases caused by Phytomonas spp. The study design is fine and the results are convincing.

The following are specific remarks:

Abstract

Line 23, ROS, should be given full spellings

Line 43, replace “Phytomonas staheli” by “P. staheli”

Line 44, replace “Phytomonas leptovasorum” by “P. leptovasorum”

Line 45, replace “The specie” by “The species”

Line 65, replace “ . “ by “ , “

Materials and Methods

Line 101, “To determine de”. replace by “To determine the” ??

Lines 108, 110, 25ºC. 80ºCfor, check and correct.

Line 144, M & M and Figs. 1,2,4; (p* <0.05, p** <0.01….. ) replace by (*p<0.05, **p<0.01…..)

Results

Line 155, Figs. 1; (p* <0.05, p** <0.01….. ) replace by (*p<0.05, **p<0.01…..), including figs. 2,4

Discussion

Line 217, replace “Phytomonas serpens” by “P. serpens”

Line 245-246, replace “Leishmania amazonensis” by “L. (Leishmania) amazonensis”

Lines 262, 268; “Phytomonas” should be in italic

Line 290, replace “L. mexicana” by “L. (L.) mexicana

References:

Check and correct all the Scientific names throughout the section, e.g. Phytomonas Serpens; Leishmania Amazonensis; Leishmania Braziiliensis, Trypanosoma Cruzi, and so on …., including the style of citation/description, following the journal instruction.

Author Response

The study is an extension of some previous work of the authors on the activity of chalconess and delivatives in Phytomonas spp., where they first evaluated that the compound (E)-1-phenyl-3-αnaphthylprop-2-en-1-one (NaF) is the most active. This study reports the results of further investigated NaF mode of action, analyzing the effect by ultrastructural and fluorimetric analysis. The result presents an initial overview of the action mode of chalcone derivative NaF that is easily synthetized and shows a strong antiproliferative effect on the promastigotes of P. serpens; suggesting a good start point for the development of a new treatment to control plant diseases caused by Phytomonas spp. The study design is fine and the results are convincing.

The following are specific remarks:

Abstract

Line 23, ROS, should be given full spellings

Line 43, replace “Phytomonas staheli” by “P. staheli”

Line 44, replace “Phytomonas leptovasorum” by “P. leptovasorum”

Line 45, replace “The specie” by “The species”

Line 65, replace “ . “ by “ , “

 Materials and Methods

Line 101, “To determine de”. replace by “To determine the” ??

Lines 108, 110, 25ºC. 80ºCfor, check and correct.

Line 144, M & M and Figs. 1,2,4; (p* <0.05, p** <0.01….. ) replace by (*p<0.05, **p<0.01…..)

 Results

Line 155, Figs. 1; (p* <0.05, p** <0.01….. ) replace by (*p<0.05, **p<0.01…..), including figs. 2,4

 Discussion

Line 217, replace “Phytomonas serpens” by “P. serpens”

Line 245-246, replace “Leishmania amazonensis” by “L. (Leishmania) amazonensis”

Lines 262, 268; “Phytomonas” should be in italic

Line 290, replace “L. mexicana” by “L. (L.) mexicana

 References:

Check and correct all the Scientific names throughout the section, e.g. Phytomonas Serpens; Leishmania Amazonensis; Leishmania Braziiliensis, Trypanosoma Cruzi, and so on …., including the style of citation/description, following the journal instruction.

 Answer: All mistakes were corrected. All references were revised and followed by MDPI guide to authors.